# Retrospective Monocentric Clinical Study on Male Infertility: Comparison between Two Different Therapeutic Schemes Using Follicle-Stimulating Hormone

**DOI:** 10.3390/jcm10122665

**Published:** 2021-06-17

**Authors:** Rosita A. Condorelli, Rossella Cannarella, Andrea Crafa, Federica Barbagallo, Laura M. Mongioì, Antonio Aversa, Emanuela Greco, Aldo E. Calogero, Sandro La Vignera

**Affiliations:** 1Department of Clinical and Experimental Medicine, University of Catania, 95123 Catania, Italy; rosita.condorelli@unict.it (R.A.C.); crafa.andrea@outlook.it (A.C.); federica.barbagallo11@gmail.com (F.B.); lauramongioi@hotmail.it (L.M.M.); acaloger@unict.it (A.E.C.); sandrolavignera@unict.it (S.L.V.); 2Department of Experimental and Clinical Medicine, University Magna Graecia of Catanzaro, 88100 Catanzaro, Italy; aversa@unicz.it; 3Department of Health Sciences, University Magna Graecia of Catanzaro, 88100 Catanzaro, Italy; emanuela.greco@unicz.it

**Keywords:** FSH administration, oligoasthenozoospermia, idiopathic male infertility, highly purified follicle-stimulating hormone, 75 IU daily, 150 UI three times a week

## Abstract

Follicle-stimulating hormone (FSH) is a therapeutic option in patients with idiopathic oligozoospermia and normal FSH serum levels. However, few studies have evaluated which dose of FSH is more effective. The aim of this study was to compare the clinical efficacy of the two most frequently used FSH treatment regimens: 75 IU daily vs. 150 IU three times a week. Patients were retrospectively assigned to two groups. The first group (*n* = 24) was prescribed highly purified FSH (hpFSH) 75 IU/daily (Group A), and the second group (*n* = 24) was prescribed hpFSH 150 IU three times a week (Group B) for three months. Before and after treatment, each patient underwent semen analysis, evaluation of the percentage of DNA-fragmented spermatozoa, assessment of testicular volume (by ultrasonography), and measurement of FSH and total testosterone (TT) serum levels. Treatment with hpFSH significantly improved conventional sperm parameters. In detail, sperm concentration increased significantly after treatment only in Group A, whereas total sperm count, percentage of spermatozoa with progressive motility, normal morphology, or alive improved significantly in both groups. Interestingly, the percentage of sperm DNA fragmentation decreased significantly in both groups after treatment with hpFSH. FSH serum levels were expectably higher at the end of the treatment than before hpFSH was administered to both groups. Remarkably, TT serum levels only increased significantly in Group A. Finally, testicular volume was significantly higher in Group A after treatment, while it did not change significantly compared to baseline in Group B. The percentage of FSH responders did not differ significantly between the two groups (8/24 vs. 6/24). The daily administration of hpFSH 75 IU seems more effective than using 150 IU three times a week. However, this therapeutic scheme implies a higher number of injections and slightly higher costs.

## 1. Introduction

Follicle-stimulating hormone (FSH) can be prescribed to infertile patients with oligozoospermia and serum FSH within the normal range (<8 mUI mL^−1^), given its physiological role in spermatogenesis. It can be extracted from the urine of postmenopausal women (highly purified FSH, hpFSH) or synthesized by recombinant technology (rhFSH) [1,2].

No difference has been reported between these two formulations (hpFSH vs. rhFSH) in terms of clinical efficacy [3].

Moreover, a molecular and proteomic approach has shown different molecular effects using rhFSH (α-follitropin and β-follitropin) and hpFSH [4]. On the basis of the proteomic analysis, the authors concluded that α-follitropin can be used in hypospermatogenesis due to insufficient hypogonadotropic stimulus, or to induce spermatogenesis in puberty, β-follitropin could improve spermiation or could be used in case of spermatidic arrest, and finally, urofollitropin could be useful in treating idiopathic infertility in normogonadotropic patients [4].

The administration of FSH has been shown to improve conventional sperm parameters [5,6,7]. Furthermore, the percentage of spermatozoa with fragmented DNA, in addition to the pregnancy rate, seems to ameliorate after FSH administration in patients with idiopathic infertility [8,9,10,11]. A meta-analysis of randomized studies showed that FSH treatment increases sperm concentration, total sperm count, and the percentage of spermatozoa with progressive motility. No significant effect has been reported on the percentage of normal forms [3].

Various doses of FSH and different therapeutic schemes have been used so far. Specifically, at concentrations ranging from 175 to 262.5 IU per week, FSH seems to have a significant effect only on the percentage of motile spermatozoa. A significant increase in sperm concentration and normally shaped spermatozoa was found when FSH was given at doses of 350–525 IU per week, while total sperm count and progressive motility showed a trend towards increasing. Finally, at higher doses (700–1050 IU per week), FSH increased sperm concentration, total sperm count, and progressive motility, while sperm morphology showed an upward trend [3]. Overall, these findings suggest that FSH seems to act with a dose-dependent mechanism to ameliorate spermatogenesis. However, this finding is based on a relatively small number of studies conducted in heterogeneous cohorts of patients and with varying doses of FSH and treatment durations. Most of the evidence is based on patients treated with intermediate doses of FSH, but it is unclear whether the best efficacy is achieved with daily or alternate-day administration. Indeed, treatment with hpFSH 150 IU three times a week is the most used scheme in clinical studies. To the best of our knowledge, no randomized clinical trial has used a daily administration of hpFSH 75 IU, and no study has compared these two treatment schemes so far. Therefore, the present study aims to evaluate the effects of treatment with FSH administered daily or three times a week. For this purpose, hpFSH 75 IU per day (525 IU per week) or 150 IU three times per week (450 IU per week) was prescribed to infertile patients with oligozoospermia and normal serum FSH levels according to routine clinical practice. Conventional sperm parameters, sperm DNA fragmentation rate, testicular volume, and FSH and serum total testosterone (TT) levels, evaluated before and after 3 months of FSH treatment, were then analyzed retrospectively.

## 2. Patients and Methods

### 2.1. Patient Selection

This is a retrospective monocentric clinical study performed on patients who were referred to the Division of Endocrinology, Metabolic Diseases, and Nutrition, University of Catania, for male infertility. We consecutively recruited 48 Caucasian men older than 18 years. The following exclusion criteria were established for this study: azoospermia, head injury, endocrine disorders (hypogonadism, hyperprolactinemia, Cushing syndrome, acromegaly, hypopituitarism), low or high (>8.0 IU mL^−1^) serum FSH levels, systemic diseases (kidney, liver, diabetes mellitus, etc.), genetic disorders, male accessory gland infection, and varicocele of high grade (>II). The following inclusion criteria were also established: sperm concentration > 5 mil/mL, mean testicular volume ranging from 8 to 12 mL, and TT > 350 ng dL^−1^. The cost analysis was performed using the prices of commercially available hpFSH (Fostimon, IBSA Farmaceutici Italia S.r.l., Lodi, Italy) vials of 75 IU (10 vials = EUR 215.38) and 150 IU (5 vials = EUR 215.38).

All patients were prescribed FSH according to the protocols regularly used in clinical practice. They were retrospectively assigned into two groups, A (*n* = 24) and B (*n* = 24). The patients of Group A were treated with hpFSH at the dose of 75 IU daily, whereas those of Group B were prescribed 150 IU three times a week. All patients were treated for three months. Before and at the end of the treatment, each patient underwent semen analysis, evaluation of the percentage of spermatozoa with fragmented DNA, evaluation of testicular volume by ultrasound, and the measurement of FSH and TT serum levels according to the routine clinical protocol used for infertile patients in our division.

### 2.2. Semen Analysis

Semen samples were collected by masturbation into a sterile container after 2–7 days of sexual abstinence and were analyzed immediately after liquefaction. According to the 2010 WHO guidelines, each sample was evaluated for seminal fluid volume, pH, sperm count, motility, morphology, and round cell concentration.

### 2.3. Sperm DNA Fragmentation

Sperm DNA fragmentation was evaluated by flow cytometry using an EPICS XL (Becker Coulter, Milan, Italy), as previously reported in [12]. DNA fragmentation was evaluated by terminal deoxynucleotidyl transferase-mediated deoxyuridine triphosphate nick-end labeling (TUNEL) staining. The negative control was obtained by not adding terminal deoxynucleotidyl transferase to the reaction mix, while the positive control was obtained by pretreating spermatozoa with 1 mg/mL of RNase-free deoxyribonuclease I (Sigma Chemical, St. Louis, MO, USA) at 37 °C for 60 min before labeling.

### 2.4. Scrotal Ultrasound Evaluation

The ultrasound examination was performed with a GX Megas Esaote (Esaote SpA, Genoa, Italy) equipped with a linear, high-resolution, and high-frequency (10 MHz) probe. This probe is suitable for the study of soft areas of the body, with echo color Doppler for slow flow detection and a scanning surface of at least 5 cm. Testicular volume was calculated using the ellipsoid formula (length × width × thickness × 0.52). The testis was considered normal in size when it had a volume between 15 and 25 cm^3^, low normal when it had a volume between 10 and 12 cm^3^, and hypotrophic when it had a volume lower than 10 cm^3^ [13,14]. Testicular volume was evaluated by adding the volumes of the right and left testes.

### 2.5. Hormonal Measurements

The hormone evaluation was performed by electrochemiluminescence (ECLIA) (Roche Cobas, Mannheim, Germany). The reference values were 0.95–11.95 IU L^−1^ for FSH and 300–1000 ng dL^−1^ for TT. Blood was collected from all patients at 08.00 in the morning.

### 2.6. Statistical Analysis

Results are reported as mean ± SD throughout the study. Values are shown as median with 95% confidence interval (CI) in figures. The normal distribution of each variable was evaluated with the Shapiro–Wilk test. Intra- and inter-group differences were evaluated by one-way analysis of variance (ANOVA). Differences in the percentage of responders (defined as patients whose total sperm count doubled after 3 months of treatment compared to before therapy between groups A and B) were evaluated by the Chi-squared test. To investigate the role of age, FSH serum levels, total sperm count, testicular volume, and percentage of sperm with fragmented DNA at enrolment, these variables were all included in a multivariate regression analysis with a stepwise procedure performed for post-treatment values of total sperm count and percentage of sperm with fragmented DNA for each group. Statistical analysis was performed using MedCalc^®^ Statistical Software version 20.006 (MedCalc Software Ltd., Ostend, Belgium; https://www.medcalc.org; accessed on 17 June 2021). (Version 19.6–64 bit). A *p*-value lower than 0.05 was accepted as statistically significant.

### 2.7. Ethical Approval

The protocol was approved by the Internal Institutional Review Committee. Written informed consent was obtained from each participant after a full explanation of the purpose of the data collection. The diagnostic–therapeutic procedure used was that routinely used in infertile patients according to the main guidelines [15]. The study was conducted according to the principles of the Declaration of Helsinki.

## 3. Results

Group A and Group B did not differ significantly in age, body mass index (BMI), testicular volume, FSH, or TT serum levels (Table 1). No difference was found in baseline conventional sperm parameters or the percentage of DNA-fragmented spermatozoa between the groups of patients (Figure 1 and Figure 2).

Treatment with FSH significantly improved conventional sperm parameters. In particular, sperm concentration (Figure 1, panel A) had significantly increased by the end of therapy only in Group A, whereas total sperm count (Figure 1, panel B), the percentage of spermatozoa with progressive motility (Figure 1, panel C), normal morphology (Figure 1, panel D) and alive (Figure 1, panel E) significantly improved in both groups. Interestingly, the percentage of spermatozoa with fragmented DNA had significantly decreased in both groups by the end of the treatment (Figure 2).

FSH serum levels were significantly higher at the end of therapy compared to baseline values in both groups (Figure 2, panel A) and, remarkably, TT levels increased significantly only in Group A (Figure 2, panel B). Finally, testicular volume was significantly higher in Group A at the end of therapy, whereas it did not change in Group B (Figure 2, panel C).

The percentage of FSH responders did not differ between Group A and Group B (8/24 vs. 6/24).

To better evaluate the role, if any, of age, BMI, FSH, TT, testicular volume, total sperm count, and the percentage of DNA-fragmented spermatozoa before treatment on total sperm count and sperm DNA fragmentation after treatment, we built a multivariate regression model including all these variables. None of them predicted the improvement of sperm DNA fragmentation in both groups and total sperm count in Group A. Total sperm count before therapy was the only variable able to predict the increase in total sperm count at the end of treatment, but only in Group B (Appendix A).

The result of the cost analysis is shown in Figure 3. It shows a higher cost of the therapeutic scheme prescribed to the patients of Group A compared to those of Group B.

## 4. Discussion

The use of FSH for the treatment of infertile patients with oligozoospermia and serum FSH within the normal range represents a valid therapeutic strategy supported by scientific evidence [15]. The concomitant presence of some *FSHβ* or *FSHR* gene polymorphisms, the arrest of spermatogenesis at the stage of spermatid, and high baseline FSH serum levels, especially if associated with low inhibin B serum levels and low testicular volume, can compromise the response to FSH treatment (poor FSH responders) [15].

The results of this study confirmed that FSH administration is capable of improving conventional sperm parameters when given at weekly doses ranging from 450–525 IU. Several studies have shown that FSH treatment increases the spermatogonial population and sperm count in oligozoospermic patients [5,16,17]; others have shown increased fertilization and pregnancy rates [18,19]. Indeed, FSH has a pivotal role in testicular development and spermatogenesis, especially in the initial mitotic phase, by binding to a specific receptor located on the surface of the Sertoli cells that are important for these physiological processes. In vitro studies also confirm this aspect. Indeed, highly purified pre-pubertal Sertoli porcine, incubated for 48 h with FSH or FSH and testosterone, showed the presence of Sertolian extracellular vesicles containing specific mRNAs [20]. In detail, the proteomic analysis identified 29 proteins related to receptor binding activity. FSH-stimulation-induced increases in inhibin-α, inhibin-β, plakoglobin, haptoglobin, d-3-phosphoglycerate dehydrogenase and sodium/potassium-transporting ATPase. Testosterone stimulation enhanced the abundance of inhibin-α, inhibin-β, tissue-type plasminogen activator, and epidermal growth factor-like protein 8, elongating factor 1-gamma and d-3-phosphoglycerate dehydrogenase. These results confirmed the presence of unknown molecular secretions of Sertoli cells [20].

FSH increased significantly total sperm count and sperm progressive motility, morphology, and vitality in both groups. The FSH posology of 75 IU/daily (Group A) also improved sperm concentration compared to an alternate day posology (Group B). These data could be due to the daily administration of FSH and, therefore, to the constant action of FSH on the mitotic phase of spermatogenesis. This could increase the pool of spermatogonia that will differentiate into spermatozoa; however, to date, there are no studies that can support this hypothesis.

Our results confirmed the protective role of FSH administration on sperm DNA fragmentation. DNA double-strand breaks can occur during spermatogenesis or sperm maturation and epididymal transit [21,22,23]. Sperm DNA fragmentation rate has been proposed as an end point to evaluate the efficacy of treatment with FSH in male infertility [24].

Two meta-analyses have shown the beneficial effect of FSH administration in male infertility in terms of pregnancy rates in couples who undergo assisted reproductive techniques [10,25]. Moreover, an elevated percentage of DNA-fragmented spermatozoa decreases the pregnancy rate and increases the recurrent miscarriage rate [26]. Therefore, these findings strongly suggest that FSH treatment improves the pregnancy rate in couples including men with idiopathic infertility by reducing sperm DNA fragmentation. Moreover, FSH may also be prescribed in patients with increased sperm DNA fragmentation resulting from several conditions, such as male age, elevated BMI, cigarette smoke, etc., especially when associated with increased oxidative stress in the seminal fluid [27].

As expected, FSH serum levels had increased by the same degree in both groups at the end of therapy. Interestingly, TT serum levels were higher in Group A compared to Group B at the end of treatment. Testicular volume increased significantly only in the patients of Group A, but not in those of Group B by the end of the treatment. These findings, taken together with the increased TT serum levels and sperm concentration, suggest that daily administration of FSH may exert greater therapeutic effects than the traditional scheme of 150 IU three times a week. In support of these findings, FSH, in addition to its well-established effects on the testicular function on spermatogenesis, exerts an intratesticular control on androgen production [12]. Local intratesticular regulators can act in a paracrine and/or autocrine fashion. Hence, molecules synthesized in one cell bind to membrane receptors present in the same or neighboring cells and regulate their function. Data from several experimental approaches clearly indicate the existence of multiple interactions between testicular cells and the potential role of these interactions in the paracrine control of testicular functions. Sertoli cells play an important role in spermatogenesis and one of their functions is the production of substances with endocrine or paracrine effects, not only to regulate spermatogenesis, but also to interact with interstitial testosterone-secreting Leydig cells [28]. These effects seem to be mediated mainly by diffusible steroidogenic factors, enhanced by FSH treatment, that control the multiplication, meiotic reduction, and maturation of germ cells [29]. These data are further supported by in vitro evidence showing that co-cultures of pig Leydig cells with homologous or heterologous Sertoli cells enhance Leydig cells’ specific functions and cause their hypertrophy for 2–3 days. Therefore, Sertoli and Leydig cells seem to reciprocally influence each other, and this evidence supports our results.

Meta-analysis studies have shown a dose–response effect of FSH on conventional sperm parameters. This finding suggests that the administration of higher doses of FSH contributes to improving sperm quality [3] and favors an increase in testicular volume [12]. Accordingly, the administration of 75 IU daily, which leads to a slight increase in the weekly dose administered (525 vs. 450 IU), could represent a possibly more effective choice. Further studies should evaluate and compare the effects of higher FSH doses in infertile patients with oligozoospermia and normal FSH serum levels.

The cost of daily administration of FSH 75 IU (Group A) is higher than that of using three vials a week though at higher doses (Group B), at least in Italy. Although this difference seems minimal, it may become relevant and perhaps non-sustainable for a treatment that has to be given for at least three months, if not for a longer duration. Moreover, daily administration implies a greater number of injections that may impact the patients’ compliance.

The 75 IU daily hpFSH therapeutic regimen could represent an alternative and perhaps better strategy for infertile patients with a higher risk of developing hypogonadism in the medium term. Although no definitive explanation can be provided for our findings, two hypotheses may be proposed: (1) the 75 IU daily scheme may provide greater cumulative exposure to FSH in the three months of treatment (6300 vs. 5400 IU; equivalent to about 14.3% higher dose) compared to 150 IU three times a week; (2) daily administration can offer more physiological FSH supplementation than that of three times a week. Both these aspects may play an important role in the understanding of the present findings.

## 5. Conclusions

This study further confirms the efficacy of the treatment with FSH in infertile patients with oligozoospermia and FSH serum levels within the normal range. In addition to the current knowledge, these results highlight the possibility for a treatment schedule different from the traditional one (150 IU three times a week).

## Figures and Tables

**Figure 1 jcm-10-02665-f001:**
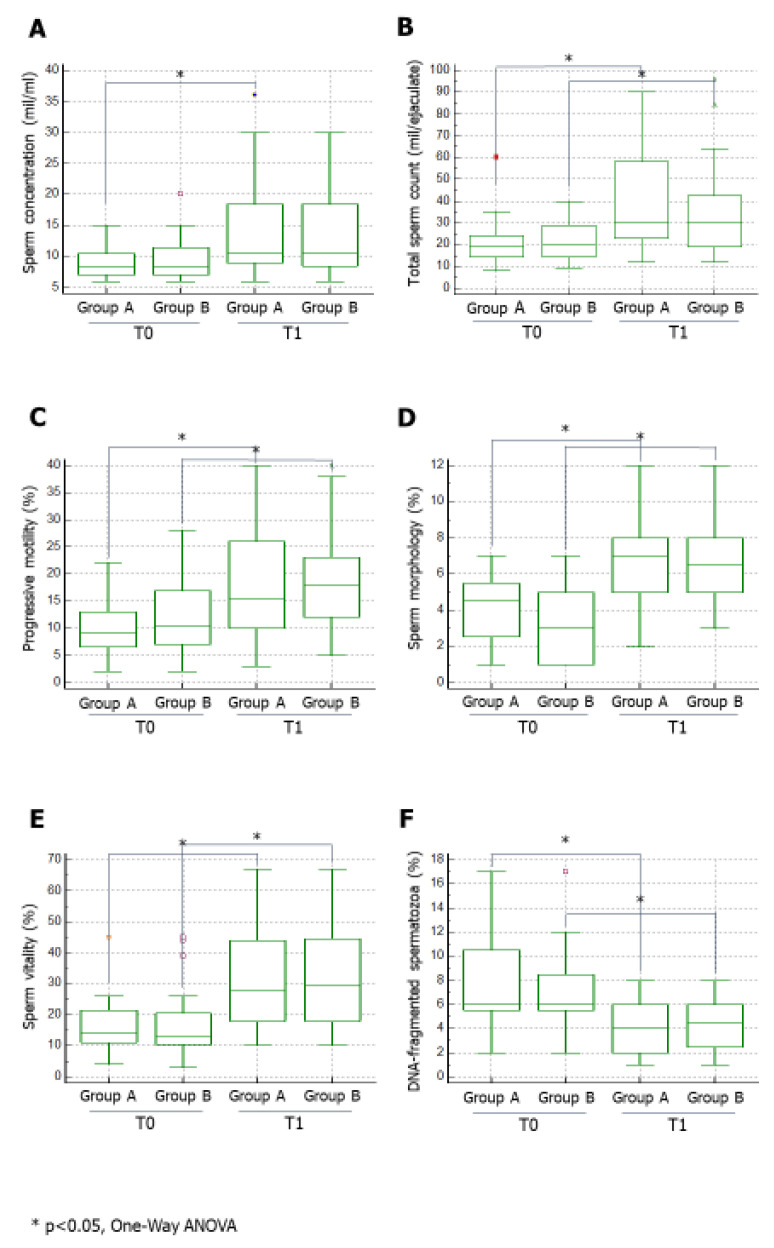
Effects of the two FSH therapeutic schemes on conventional sperm parameters. Data on sperm concentration (panel **A**), total sperm count (panel **B**), sperm motility (panel **C**), sperm morphology (panel **D**) and sperm vitality (panel **E**) and percentage of spermatozoa with fragmented DNA (panel **F**) are shown. FSH, follicle-stimulating hormone. * *p* < 0.05.

**Figure 2 jcm-10-02665-f002:**
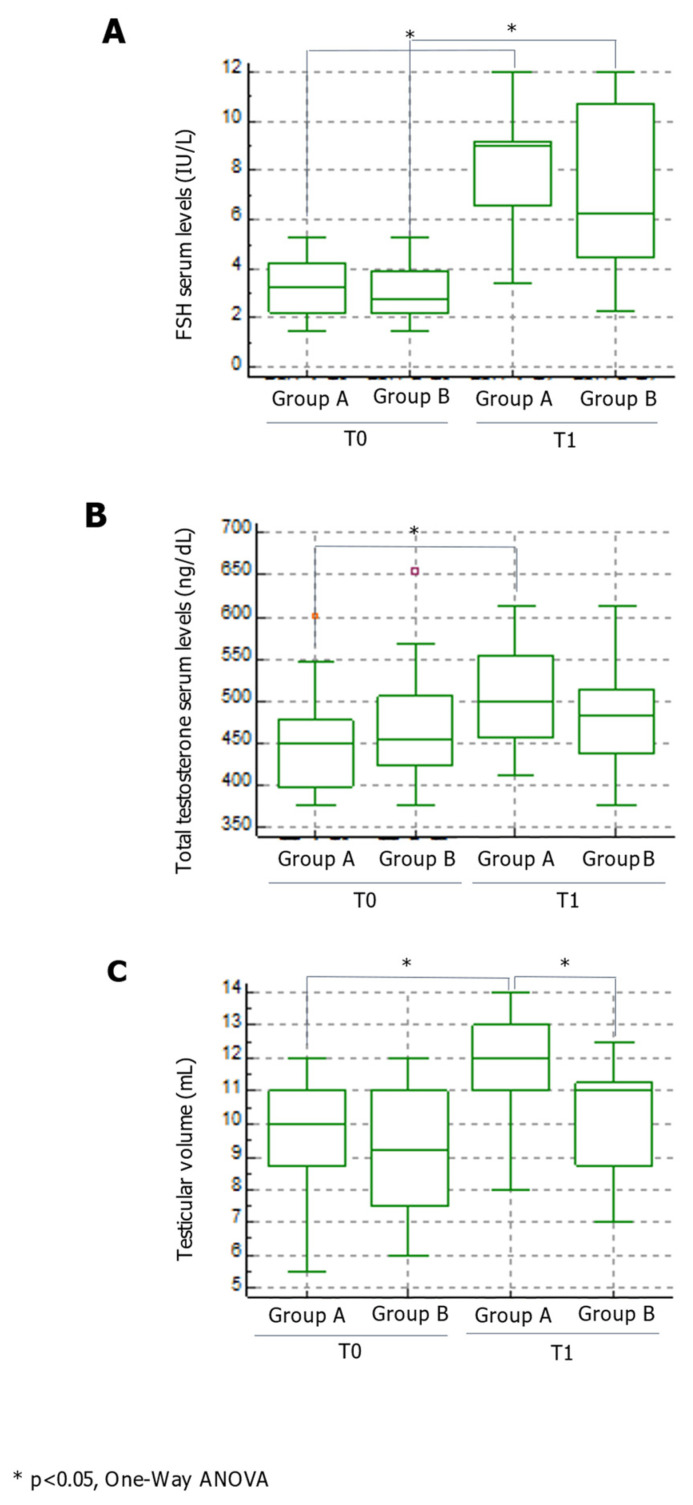
Effects of the two FSH therapeutic schemes on hormone serum levels and testicular volume. Data on FSH (panel **A**) and TT (panel **B**) serum levels, and on testicular volume (panel **C**) are shown. FSH, follicle-stimulating hormone; TT, total testosterone. * *p* < 0.05.

**Figure 3 jcm-10-02665-f003:**
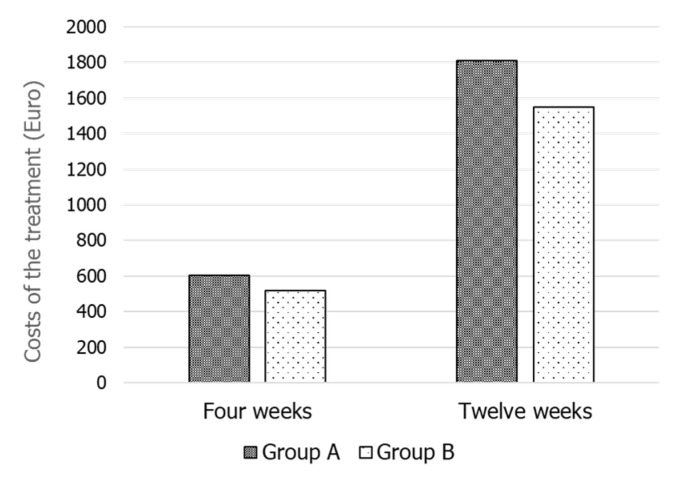
Cost analysis of the two FSH therapeutic schemes for 4 or 12 weeks of treatment.

**Table 1 jcm-10-02665-t001:** Baseline parameters.

	Group A (*n* = 24)	Group B (*n* = 24)
Age (years)	33.8 ± 6.1	34.8 ± 5.2
Body mass index (kg m^2 −1^)	25.5 ± 2.3	24.3 ± 2.8
Follicle-stimulating hormone (IU L^−1^)	3.1 ± 1.2	3.0 ± 1.1
Total testosterone (ng L^−1^)	452.1 ± 58.4	466.2 ± 65.1
Testicular volume (mL)	9.8 ± 1.6	9.2 ± 2.1
Sperm concentration (mil mL^−1^)	8.8 ± 2.4	9.5 ± 3.2
Total sperm count (mil ejaculate^−1^)	20.6 ± 10.5	21.6 ± 9.0
Sperm progressive motility (%)	10.1 ± 5.8	11.6 ± 6.3
Sperm morphology (%)	4.2 ± 2.0	3.2 ± 2.1
Sperm vitality (%)	15.7 ± 8.7	16.6 ± 11.6

Data are expressed as mean ± standard deviation.

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
