# Peer review of "Retrospective Monocentric Clinical Study on Male Infertility: Comparison between Two Different Therapeutic Schemes Using Follicle-Stimulating Hormone"

_jcm, 2021, doi:10.3390/jcm10122665_

Round 1

Reviewer 1 Report

Condorelli et al. performed their study comparing the effect on sperm parameters of the administration of daily 75 UI and of 150 UI x 3/week of hpFSH.  The subject is of interest and the manuscript is well written.

In my opinion, minor revisions should be performed before publication.

  • Line 43: Although no differences have been reported between hpFSH and rhFSH, some in vitro observations have been reported about the different effect of different FSH preparations on Sertoli cells (Arato I et al, 2020; doi: 10.3389/fendo.2020.00401.), thus suggesting the preferential use of hpFSH in patients with idiopathic oligozoospermia (as done by Condorelli et al in this paper). Authors should discuss this aspect in the introduction.
  • Patients and methods: If partial maturative disturbances have not been excluded by testicular fine needle aspiration, before the treatment, this aspect should be reported among the limits of the study. Furthermore the absence of stratification of patients for FSHbeta or FSHR gene polymorphisms should be reported among the limits of the study.
  • Results: It would be interesting that authors may provide informations about how many patients significantly increased seminal parameters (responders) and how many patients may be considered as "non-responders", if there are significant differences in the 2 groups, and to compare seminal parameters not only in the 2 populations (A and B group) as a whole, but moreover in the 2 sub-groups of responders, thus excluding non-responders patients.
  • Line 236: we suggest to comment more in details the importance of FSH in the molecular machinery involved in spermatogenesis (Mancuso F et al 2018, DOI: 10.1016/j.mce.2018.04.001)

Author Response

Answers to Reviewer#1 comments

Manuscript ID jcm-1250289

The introduction and the English language have been improved as required.

Comment 1: Line 43: Although no differences have been reported between hpFSH and rhFSH, some in vitro observations have been reported about the different effect of different FSH preparations on Sertoli cells (Arato I et al, 2020; doi: 10.3389/fendo.2020.00401.), thus suggesting the preferential use of hpFSH in patients with idiopathic oligozoospermia (as done by Condorelli et al in this paper). Authors should discuss this aspect in the introduction.

Answer to comment 1:

Thanks for the valuable comment. We have enriched the introduction with the study you suggested. The changes are highlighted in yellow in the manuscript.

Comment 2: Patients and methods: If partial maturative disturbances have not been excluded by testicular fine needle aspiration, before the treatment, this aspect should be reported among the limits of the study. Furthermore the absence of stratification of patients for FSHbeta or FSHR gene polymorphisms should be reported among the limits of the study.

Answer to comment 2:

This observation has been considered and therefore we have added a paragraph with the limitations of the study. These data are missing and not available as it is a retrospective study The changes are highlighted in yellow in the manuscript.

Comment 3: Results: It would be interesting that authors may provide informations about how many patients significantly increased seminal parameters (responders) and how many patients may be considered as "non-responders", if there are significant differences in the 2 groups, and to compare seminal parameters not only in the 2 populations (A and B group) as a whole, but moreover in the 2 sub-groups of responders, thus excluding non-responders patients.

Answer to comment 3:

Thank you for your valuable comment.

Differences in the percentage of responders [defined as patients whose total sperm count doubled after 3 months of treatment compared to before therapy (Ferlin et al., 2011)] between groups A and B were evaluated by the Chi-squared test (please see page 3, section “Statistical analysis”). The percentage of FSH-responders did not differ between Group A and Group B (8/24 vs. 6/24) (lines 174-175). No significant difference in seminal parameters was found between the responders of group A and of group B, which might be ascribed to the low number of patients (8 vs. 6).

Comment 4: Line 236: we suggest to comment more in details the importance of FSH in the molecular machinery involved in spermatogenesis (Mancuso F et al 2018, DOI: 10.1016/j.mce.2018.04.001)

Answer to comment 4:

We accepted your suggestion and with it we have enriched our discussion. The changes are highlighted in yellow in the manuscript.

Reviewer 2 Report

Condorelli and colleagues present a manuscript entitled “Retrospective monocentric clinical study on male infertility: Comparison between two different therapeutic schemes using follicle-stimulating hormone.” The study is interesting in that it compares the effectiveness of two different FSH-based treatment regimens for idiopathic oligozoospermia in men. Semen parameters were improved in both groups, and a 75 IU FSH/day regimen did appear to yield better treatment outcomes. However, additional experimental detail is required and the multiple variables between the two treatment groups make it difficult to ascertain the biological reason for the difference in outcomes. Specific concerns are listed below.

  1. The two FSH treatment regimens differed both in terms of periodicity of administration and total weekly dosage. While it is recognized that comparison of two treatment regimens is the purpose of the study, incorporating two different variables at once renders comparison of the variables problematic. Was it the increased number of administrations per week, increased dosage per week, or both that improved outcomes? This is also important in terms of cost-effectiveness.
  2. The authors provide an analysis of cost-effectiveness for the two FSH treatment regimens. A graph comparing the actual cost for four weeks and twelve weeks may not be necessary. The cost could be stated in the text since the cost analysis appears to have only been performed using the cost for the medication. Does this cost analysis assume that the FSH injections would be self-administered? In other locations, injections may occur in the clinic, which would significantly impact the cost effectiveness.
  3. The authors state that the number of FSH-responders was 8/24 and 6/24 in the two treatment groups. How were FSH-responders identified? Are only the data for the FSH-responders shown in the graphs, or do the graphs illustrate the results for all study participants?
  4. In this study, men treated daily with 75 IU FSH were found to have a statistically significant higher total serum testosterone (TT) than men treated every 3 days with 150 IU FSH. However, all TT levels, including the elevated TT in the daily treated group, were within the normal range for TT. TT can vary with time of day, and blood draws for TT are recommended to be taken in the morning to avoid variability. TT is typically highest in the morning compared to later in the day. Do the authors know if blood draws were obtained consistently at the same time in both treatment groups? Interestingly, a study that used a similar treatment regimen did not observe an increase in TT (Ben-Rafael et al. 2000 Fertility and Sterility). By way of explanation for their surprising finding regarding TT, the authors rightly discuss the paracrine effects that Sertoli cells can have on Leydig cell function. They also cite a previous study from their own group that indicates a positive effect of FSH treatment on TT; however, in this previous study FSH treatment appeared to prevent a decrease in TT, rather than cause an increase in TT.
  5. The authors propose two hypotheses based on their results in the Conclusions section. These speculations should be elaborated upon in the Discussion and not just mentioned in the Conclusions.
  6. The term “sperm analysis” in this manuscript is more commonly referred to as semen analysis.

Author Response

Answers to Reviewer#2 comments

Manuscript ID jcm-1250289

The introduction, methods, conclusions and the English language have been improved as required.

Comment 1: The two FSH treatment regimens differed both in terms of periodicity of administration and total weekly dosage. While it is recognized that comparison of two treatment regimens is the purpose of the study, incorporating two different variables at once renders comparison of the variables problematic. Was it the increased number of administrations per week, increased dosage per week, or both that improved outcomes? This is also important in terms of cost-effectiveness.

Answer to comment 1:

Thanks for your comment and I share your observations. The doses administered in this study refer to the two dosages most described in the scientific literature in cases of treatment with FSH in idiopathic normogonadotropic infertile patients. The choice and therefore the prescription of these two treatment schemes respect that described in the literature illustrating a retrospective clinical experience. Therefore, the purpose of the study was to understand the real effectiveness of the two proposed schemes in terms of cost benefit. This our observation is reported in the text: “Although no definitive explanation can be provided for our findings, two hypotheses may be proposed: 1) the 75 IU daily scheme may provide greater cumulative exposure to FSH in the three months of treatment (6300 vs. 5400 IU; equivalent to about 14.3% higher dose) compared to 150 IU three times a week; 2) the daily administration can offer a more physiological FSH supplementation than that three times a week.  Both these aspects may play an important role in the understanding of the present findings.”

This is the first study comparing the two treatment schemes and therefore future studies expanding to other endpoints could better clarify the real advantage of daily therapy against a higher cost.

Comment 2: The authors provide an analysis of cost-effectiveness for the two FSH treatment regimens. A graph comparing the actual cost for four weeks and twelve weeks may not be necessary. The cost could be stated in the text since the cost analysis appears to have only been performed using the cost for the medication. Does this cost analysis assume that the FSH injections would be self-administered? In other locations, injections may occur in the clinic, which would significantly impact the cost effectiveness.

Answer to comment 2:

We preferred to indicate the cost analysis for greater accuracy in the discussion of the results. Of course, in the face of a benefit that may seem minimal even if in a category of "idiopathic" patients where the treatment is always difficult for the clinician, administering a dose of 75 IU/ day seems to be the best solution. The costs indicated refer to the number of vials self-administered but it is clear that in other situations or locations the cost could have a greater impact. To date, there is no consensus on the scheme to be used and clinicians use more the administration of 150 IU three times a week on the basis of the only fact that it is the most used scheme. Therefore, we believe that comparative studies, such as this, can be useful to broaden and better clarify this aspect.

Comment 3: The authors state that the number of FSH-responders was 8/24 and 6/24 in the two treatment groups. How were FSH-responders identified? Are only the data for the FSH-responders shown in the graphs, or do the graphs illustrate the results for all study participants?

Answer to comment 3: Responders were defined as patients whose total sperm count doubled after 3 months of treatment compared to before therapy (Ferlin et al., 2011)] (please see page 3, section “Statistical analysis”). The graphs indicate the data for all the study participants.

Comment 4: In this study, men treated daily with 75 IU FSH were found to have a statistically significant higher total serum testosterone (TT) than men treated every 3 days with 150 IU FSH. However, all TT levels, including the elevated TT in the daily treated group, were within the normal range for TT. TT can vary with time of day, and blood draws for TT are recommended to be taken in the morning to avoid variability. TT is typically highest in the morning compared to later in the day. Do the authors know if blood draws were obtained consistently at the same time in both treatment groups? Interestingly, a study that used a similar treatment regimen did not observe an increase in TT (Ben-Rafael et al. 2000 Fertility and Sterility). By way of explanation for their surprising finding regarding TT, the authors rightly discuss the paracrine effects that Sertoli cells can have on Leydig cell function. They also cite a previous study from their own group that indicates a positive effect of FSH treatment on TT; however, in this previous study FSH treatment appeared to prevent a decrease in TT, rather than cause an increase in TT.

Answer to comment 4:

Blood sampling for TT was performed at the same time slot for all patients. We have clarified this aspect in the methods. The comparison with our previous study published in 2019 (PMID: 31240459 DOI: 10.1007/s12020-019-01983-0) is not easy, because they are two different study designs. The first compared patients treated with FSH versus untreated patients, 4 and 12 months after the end of a treatment period that lasted 4 months. However, we believe that the following limitations of the study "absence of testicular fine needle aspiration, performed before of the treatment and the absence of stratification of patients for FSHβ or FSHR gene polymorphisms", included in a specific paragraph in the revised version, will help in the future to understand the differences that also affect our results compared to those of Ben-Rafael et al. 2000 Fertility and Sterility.

Comment 5: The authors propose two hypotheses based on their results in the Conclusions section. These speculations should be elaborated upon in the Discussion and not just mentioned in the Conclusions.

Answer to comment 5:

We have expanded the discussion by inserting the hypotheses of the conclusions as you have well highlighted.

Comment 6: The term “sperm analysis” in this manuscript is more commonly referred to as semen analysis.

Answer to comment 6:

The term “semen analysis” is more correct and we have modified in the manuscript it, as you suggest. The changes are highlighted in yellow in the manuscript.

Round 2

Reviewer 2 Report

The authors have adequately addressed the concerns of this reviewer.